# Investigation of the Performance of Perovskite Solar Cells with ZnO-Covered PC_61_BM Electron Transport Layer

**DOI:** 10.3390/ma16145061

**Published:** 2023-07-18

**Authors:** Ting-Chun Chang, Chen-Yi Liao, Ching-Ting Lee, Hsin-Ying Lee

**Affiliations:** 1Department of Photonics, National Cheng Kung University, Tainan 701, Taiwan; l78111508@gs.ncku.edu.tw (T.-C.C.); t104360451@ntut.org.tw (C.-Y.L.); ctlee@ee.ncku.edu.tw (C.-T.L.); 2Department of Electrical Engineering, Yuan Ze University, Taoyuan 320, Taiwan

**Keywords:** perovskite solar cells, PC_61_BM electron transport layer, ZnO interface layer, time-resolved photoluminescence spectroscopy

## Abstract

Due to its high carrier mobility and electron transmission, the phenyl-C_61_-butyric acid methyl ester (PC_61_BM) is usually used as an electron transport layer (ETL) in perovskite solar cell (PSC) configurations. However, PC_61_BM films suffer from poor coverage on perovskite active layers because of their low solubility and weak adhesive ability. In this work, to overcome the above-mentioned shortcomings, 30 nm thick PC_61_BM ETLs with different concentrations were modeled. Using a 30 nm thick PC_61_BM ETL with a concentration of 50 mg/mL, the obtained performance values of the PSCs were as follows: an open-circuit voltage (V_oc_) of 0.87 V, a short-circuit current density (J_sc_) of 20.44 mA/cm^2^, a fill factor (FF) of 70.52%, and a power conversion efficiency (PCE) of 12.54%. However, undesired fine cracks present on the PC_61_BM surface degraded the performance of the resulting PSCs. To further improve performance, multiple different thicknesses of ZnO interface layers were deposited on the PC_61_BM ETLs to release the fine cracks using a thermal evaporator. In addition to the pavement of fine cracks, the ZnO interface layer could also function as a hole-blocking layer due to its larger highest occupied molecular orbital (HOMO) energy level. Consequently, the PCE was improved to 14.62% by inserting a 20 nm thick ZnO interface layer in the PSCs.

## 1. Introduction

In the modern era, fossil-energy-based environmental pollution caused by the processes of industrial development and production has posed serious problems worldwide. Therefore, research on clean, pollution-free, sustainable energy has drawn rapidly increasing attention [1]. Among the possible alternative energies, solar energy is undoubtedly the most valued energy source, and it might completely replace fossil energy in order to mitigate environmental pollution in the future [2]. There are several kinds of solar cells, such as organic solar cells [3,4], inorganic solar cells [5,6], perovskite solar cells (PSCs) [7,8,9,10,11,12], and so on. Nevertheless, in view of their advantages in terms of inexpensiveness, good flexibility, high carrier mobility, long carrier diffusion length, and high performance, organic–inorganic halide perovskite solar cells have become the most promising and attractive candidates for alternative energy sources [13,14].Over the past few years, the power conversion efficiency (PCE) of perovskite solar cells has increased from 3.8% to 25.5%, which is very close to the current PCE of crystalline silicon (c-Si) solar cells [15].

Recently, due to their advantages such as high carrier mobility and outstanding charge transport properties [16,17], phenyl-C_61_-butyric acid methyl ester (PC_61_BM) films have been widely used as electron transport layers (ETLs) in p-i-n perovskite solar cells. The incorporation of an appropriate ETL material will not only improve electron transportation between the cathode and the perovskite active layer but also suppress the carrier recombination in solar cells [18]. Moreover, because PC_61_BM films can be easily deposited using the spin-coating technique and a low annealing temperature, their associated manufacturing cost is significantly lower than that of comparable materials [19]. Therefore, traditional titanium dioxide (TiO_2_) ETLs have been gradually replaced by PC_61_BM ETL because the manufacture of TiO_2_ ETLs requires high temperatures for sintering [20,21]. However, due its low solubility and weak adhesive ability, PC_61_BM not only covers the perovskite active layer incompletely but also renders the resulting film easily prone to generating undesired fine surface cracks when it is coated on the perovskite active layer [22]. Several methods have been used to overcome the shortcomings of using PC_61_BM films as coatings [23,24]. Among these methods, the method consisting of the sandwiching of the zinc oxide (ZnO) interface layer not only filled the fine cracks on the surface of the PC_61_BM ETL but also ensured that the energy levels between the PC_61_BM ETL and silver (Ag) cathode were closely matched, thus reducing energy loss during carrier transportation. Moreover, the ZnO interface layer functioned as a hole-blocking layer owing to its larger highest occupied molecular orbital (HOMO) energy level [25,26,27]. In this work, to determine a suitable PC_61_BM ETL for use in perovskite solar cells, PC_61_BM solutions with various concentrations were mixed and investigated. Furthermore, ZnO interface layers of various thicknesses were deposited on the PC_61_BM ETL to study the features of the passivated fine cracks on the surface of the PC_61_BM ETL. As a result, perovskite solar cells were fabricated to investigate their related performance and compared.

## 2. Materials and Methods

### 2.1. Materials

In this work, indium-tin-oxide (ITO)-coated glass substrate, poly(3,4-ethylenedioxythiophene):poly(styrenesulfonate) (PEDOT:PSS) conductive solution (1.3–1.7 wt%), Methylammonium iodide (CH_3_NH_3_I, MAI, Uni-onward Corp., New Taipei City, Taiwan) powder (98%), and PC_61_BM powder (99.5%) were purchased from Uni-onward Corp., New Taipei City, Taiwan. Vanadium target was purchased from Admat Inc., Norristown, PA, USA. Lead iodide (PbI_2_) powder (99%) and ZnO powder (99%) were purchased from Alfa Aesar, Haverhill, MA, USA. Isobutyl alcohol (IBA) solvent (99%), dimethylsulfox (DMSO) solvent (99.9%), γ-butyrolactone (GBL) solvent (99%), and chlorobenzene (CB) solvent (99.8%) were purchased from Sigma-Aldrich, St. Louis, MI, USA.

### 2.2. Manufacture

Three-dimensional schematic configurations and a corresponding energy level diagram of the perovskite solar cell with PC_61_BM ETL and ZnO interface layer are shown in Figure 1a,b, respectively. At first, the 260 nm thick ITO-coated glass substrates were soaked in acetone, methanol, and deionized water and then cleaned using an ultrasonic cleaner for 5 min. Using a vanadium target, a 20 nm thick vanadium oxide (VO_x_) film was deposited on the ITO anode electrode as an interface modification layer (IML) along with an electron-blocking layer using a radio frequency magnetron-sputtering system. The VO_x_ IML enabled a greater degree of energy level matching between the work function of the ITO anode electrode and the highest occupied molecular orbital (HOMO) of the PEDOT:PSS hole transport layer (HTL) [28]. Next, a 50 nm thick IBA doped PEDOT:PSS (PEDOT:PSS:IBA, 1 mL:0.1 mL) HTL was spin-coated on the VO_x_ IML and annealed in a N_2_ ambient atmosphere at 120 °C for 15 min. MAI (0.395 g) and PbI_2_ (1.157 g) were mixed into DMSO solvent (1 mL) and GBL solvent (1 mL) to form a methylammonium lead iodide (CH_3_NH_3_PbI_3_, MAPbI_3_) perovskite solution. The perovskite solution was then spun on the PEDOT:PSS:IBA HTL via spin coating and annealed in a N_2_ ambient atmosphere at 90 °C for 20 min to form a 300 nm thick MAPbI_3_ active layer. Subsequently, to obtain various-concentration-developed PC_61_BM ETLs, the PC_61_BM material solutions were prepared by mixing CB (1 mL) with PC_61_BM (30, 50, and 70 mg), respectively. The various prepared PC_61_BM solutions were then spun on the MAPbI_3_ active layers and annealed in a N_2_ ambient atmosphere at 90 °C for 5 min to form a 30 nm thick PC_61_BM ETL. Finally, ZnO interface layers of various thicknesses (10, 20, and 30 nm) and a 100 nm thick Ag cathode electrode were sequentially evaporated on the PC_61_BM ETL using a thermal evaporator. In this study, the thicknesses of the individual layers of the resulting PSCs were confirmed using Alpha-Step (Alpha-Step D-300, KLA, Milpitas, CA, USA). Figure 1c shows the field emission scanning electron microscopy (FE-SEM, AURIGA, ZEISS, Oberkochen, Germany) cross-section images of the PSCs with a 20 nm thick ZnO interface layer. The measurement results of the thicknesses of the individual layers in the PSCs from the SEM cross-section images corresponded to the measurement results obtained using alpha step. Accordingly, the accuracy of the thicknesses of the resulting PSCs could be verified using the two different kinds of analysis mentioned above.

In this study, each parameter with various PC_61_BM concentrations and various ZnO thicknesses of the PSCs had been manufactured for over five rounds. Six pieces of the PSC samples were constructed each round, and six independent devices were resided on each piece of the PSC samples. In total, there were over 180 devices for each fabrication condition for the PSCs. Moreover, every PSC was measured to confirm the fabrication parameters. Furthermore, the yield rate of our fabrication was around 95%; thus, most of the PSCs we produced under the same parameters had similar performance. The statistical data also illustrated the good reproducibility of our fabrication parameters.

The optical transmission of the various-concentration-formed PC_61_BM ETLs was measured using a UV–Visible–NIR spectrophotometer (U-4100, HITACHI, Tokyo, Japan). The surface morphologies of the PC_61_BM and ZnO films were observed using FE-SEM. The current density–voltage (J-V) characteristics of the various perovskite solar cells were measured using a Keithley 2400 (Keithley Instruments, Cleveland, OH, USA) under an AM1.5G solar simulator (100 mW/cm^2^) (Forter Technology Corp., Taichung, Taiwan). The external quantum efficiency (EQE) spectra of the various perovskite solar cells were measured using an Xe lamp source with 150 W of power and a monochromator (QE-3000, Zolix, Beijing, China).

## 3. Results

Figure 2 provides the SEM images of the MAPbI_3_ films and the various-concentration-formed PC_61_BM ETLs/MAPbI_3_ active layers. As shown in Figure 2a, the MAPbI_3_ active layer spun on the PEDOT:PSS:IBA HTL exhibited a uniform surface. According to the SEM image shown in Figure 2b, since only some PC_61_BM material regions were observed, it could be deduced that the PC_61_BM ETL with a concentration of 30 mg/mL did not fully cover the MAPbI_3_ active layer owing to its poor adhesion. All the area that circled by red line in Figure 2b represents the covered area of PC_61_BM material as the PC_61_BM concentration of 30 mg/mL. In Figure 2c, when the concentration of the PC_61_BM material increased to 50 mg/mL, the PC_61_BM ETL uniformly and completely covered the MAPbI_3_ active layer. However, upon further increasing the concentration of the PC_61_BM material to 70 mg/mL, many cracks and pinholes were observed, as shown in Figure 2d. In general, due to the strong Van der Waals forces between the molecules of the PC_61_BM fullerene derivative, the distance between the PC_61_BM molecules decreased with an increase in the PC_61_BM concentration [29]. Therefore, the probability of the aggregation of the PC_61_BM molecules was further enhanced [30]. Consequently, cracks and pinholes were easily generated and clearly observable on the surface of the PC_61_BM ETL with a concentration of 70 mg/mL. According to the SEM images shown in Figure 2, the surface morphology of the PC_61_BM ETL was seriously affected by its own concentration. Due to the uniform coverage over the MAPbI_3_ active layer and the lack of cracks and pinholes, it was deduced that 50 mg/mL was the optimal concentration of the PC_61_BM ETL.

Photoluminescence (PL) spectroscopy using a He-Cd laser source with a wavelength of 325 nm (Kimmon Koha Corp., Tokyo, Japan) and time-resolved photoluminescence (TRPL) spectroscopy using a laser diode source with a wavelength of 375 nm (LDH-P-C 375, PicoQuant, Berlin, Germany) were used to explore the carrier recombination and the separation in the boundary between the MAPbI_3_ active layer and the PC_61_BM ETL. Figure 3a shows the PL spectra of the MAPbI_3_ active layer and the various-concentration-formed PC_61_BM ETLs/MAPbI_3_ active layers. As shown in Figure 3a, the MAPbI_3_ active layer presented the strongest PL peak intensity at the wavelength of 770 nm. On the other hand, the PL peak intensity of the resulting PC_61_BM ETLs/MAPbI_3_ active layers significantly dropped. This phenomenon was attributed to the fact that the electron–hole pairs excited using the He-Cd laser were generated in the MAPbI_3_ active layer and that the optically generated electrons could quickly transmit to the PC_61_BM ETL. Consequently, the recombination probability of electrons and holes in the MAPbI_3_ active layer was reduced, which caused the peak PL intensity of the MAPbI_3_ active layer to decrease. To confirm that the PL spectra at the wavelength of 770 nm emitted by the MAPbI_3_ active layer had not been absorbed by the PC_61_BM ETL, the transmittance values of various-concentration-formed PC_61_BM ETLs were measured; they are shown in the inset figure in Figure 3a. It was highly evident that all the PC_61_BM ETLs with various concentrations experienced highly smooth changes in transmittance from the wavelength of 300 nm to 1000 nm. The transmittance at the wavelength of 770 nm did not exhibit a significant change. This phenomenon indicated that the reduction in the peak PL intensity at the wavelength of 770 nm in the PC_61_BM ETL/MAPbI_3_ active layer was not caused by the absorption of the PC_61_BM ETL.

Figure 3b shows the TRPL spectra of the MAPbI_3_ active layer and the various-concentration-formed PC_61_BM ETLs/MAPbI_3_ active layers. The TRPL spectra were fitted using the following exponential equation (Formula (1)) [31]:(1)$$I=A\;\times \;\exp\left(-\frac{t}{\tau }\right)$$
where I denotes light intensity, A is the maximum light intensity, t is time, and τ denotes carrier lifetime. The carrier lifetime is the interval time when the light intensity decreases to 1/e of the maximum light intensity. The carrier lifetimes of the PC_61_BM ETL/MAPbI_3_ active layer with PC_61_BM concentrations of 30, 50, and 70 mg/mL were 2.00 ns, 1.51 ns, and 1.55 ns, respectively. The carrier lifetime of the MAPbI_3_ active layer was 2.41 ns, which was longer than the carrier lifetimes of all the PC_61_BM ETLs/MAPbI_3_ active layer structures. This result demonstrated that the PC_61_BM ETL could effectively transfer the electrons from the MAPbI_3_ active layer [32,33,34]. The shortest carrier lifetime of the various PC_61_BM ETLs/MAPbI_3_ active layers was yielded when a PC_61_BM concentration of 50 mg/mL was used, which was also quite well fixed the above-mentioned measurement results of the PL spectra and SEM images.

Figure 4a,b illustrate the current density–voltage (J-V) and dark current density–voltage performances of the PSCs using the various-concentration-formed PC_61_BM ETLs without ZnO interface layer covered. The open-circuit voltage (V_oc_), short-circuit current density (J_sc_), fill factor (FF), and power conversion efficiency (PCE) of the resulting PSCs are listed in Table 1. It was found that the best characteristics of the PSCs, including a V_oc_ of 0.87 V, a J_sc_ of 20.44 mA/cm^2^, an FF of 70.52%, and a PCE of 12.54%, were obtained when using the PC_61_BM ETL with a concentration of 50 mg/mL. Since the best coverage on the MAPbI_3_ surface was achieved using the PC_61_BM ETL with a concentration of 50 mg/mL, the carrier recombination probability was reduced, which resulted in the lowest dark current density. As shown in Figure 2b, since the PC_61_BM ETL with the concentration of 30 mg/mL exhibited poor adhesion, it did not completely cover the MAPbI_3_ active layer, which precluded the function of the PC_61_BM ETL and degraded the features of the MAPbI_3_ active layer due to the direct contact between the Ag cathode electrode and the MAPbI_3_ active layer [35]. Therefore, the carrier recombination rate was inevitably increased, while the electrical conductivity was reduced. These results not only reduced the J_sc_ and FF but also increased the dark current density. When the concentration of PC_61_BM was increased from 50 mg/mL to 70 mg/mL, the number of cracks increased due to the excessive aggregation of the PC_61_BM molecules, as shown in Figure 2d. The increased number of cracks increased the possibility of carrier recombination and affected electron transmission capacity. Consequently, compared to the PC_61_BM concentration of 50 mg/mL, the performance of the PSCs with a PC_61_BM concentration of 70 mg/mL was inferior. The external quantum efficiency (EQE) was an important characteristic parameter of the PSCs. Figure 4c shows the EQE and the integrated J_sc_ as a function of wavelength (300–800 nm) for the PSCs using the various-concentration-formed PC_61_BM ETLs. As shown in Figure 4c, the PSCs using the PC_61_BM ETL with a concentration of 50 mg/mL achieved the highest EQE. The trend of the EQE results also presented a significant improvement on the above-mentioned J_sc_ measurement trend of the PSCs using the various-concentration-formed PC_61_BM ETLs. This was due to the fact that the best adhesion and coverage of the PC_61_BM ETL (50 mg/mL) was achieved on the MAPbI_3_ active layer, which decreased the number of cracks and pinholes on the PC_61_BM surface and led to a reduced carrier recombination possibility and an increase in EQE. Moreover, according to the EQE results, the integrated J_sc_ values of the PSCs using the PC_61_BM ETLs with concentrations of 30, 50, and 70 mg/mL were 18.22, 19.89, and 19.00 mA/cm^2^, respectively. The values and trends of the integrated J_sc_ were all similar to the J_sc_ obtained from the J-V curve, thus verifying the J_sc_ from our perovskite solar cells and that the optimal concentration of the PC_61_BM ETL was 50 mg/mL.

Based on the above-mentioned experimental results, the best PSC performance was obtained using the PC_61_BM ETL with a concentration of 50 mg/mL. However, the PC_61_BM films still suffered from the shortcomings of low solubility and poor adhesion. To further observe the surface morphology of the PC_61_BM ETL/MAPbI_3_ active layer structure formed using 50 mg/mL, the SEM image shown in Figure 2c was extended, with the result shown in Figure 5a. It is worth noting that there were some fine cracks on the PC_61_BM surface. To overcome this problem and further improve the resulting PSCs, in this work, ZnO interface layers of various thicknesses were deposited on the PC_61_BM ETLs with a concentration of 50 mg/mL using a thermal evaporator. The morphologies of the 10, 20, and 30 nm thick ZnO interface layers/PC_61_BM ETL/MAPbI_3_ active layer structures were observed using SEM, with the resulting images shown in Figure 5b–d, respectively. As seen in the SEM image shown in Figure 5b, although the passivation function of the 10 nm thick ZnO interface layer was achieved, there were still a few areas that were not covered by the 10 nm thick ZnO interface layer. According to the images shown in Figure 5c,d, few fine cracks could be observed on the surface when the thicknesses of the ZnO interface layer were 20 nm and 30 nm. Therefore, as the thickness of the ZnO interface layer surpassed 20 nm, the fine cracks on the PC_61_BM ETL vanished completely, which was expected to enhance the carrier transportation ability and improve the performance of the resulting PSCs.

Figure 6a,b show the PL spectra and the TRPL spectra of the MAPbI_3_ active layer itself, the PC_61_BM ETL/MAPbI_3_ active layer formed using 50 mg/mL, and the 20 nm thick ZnO interface layer/PC_61_BM ETL/MAPbI_3_ active layer structures formed using 50 mg/mL, respectively. Among the PL spectra, the spectrum corresponding to the 20 nm thick ZnO interface layer exhibited the lowest PL intensity. This result was attributed to the following phenomena: the fine cracks on the PC_61_BM surface were passivated by the ZnO interface layer, and the electrons could be quickly transmitted from the MAPbI_3_ active layer and passed through the PC_61_BM ETL to the ZnO interface layer, which could reduce the carrier recombination possibility and thus lower the PL emission intensity [36,37]. Figure 6b shows that the carrier lifetime of the ZnO/PC_61_BM/MAPbI_3_ structure was 1.15 ns, which was much shorter than that of 2.41 ns for the MAPbI_3_ active layer alone and 1.51 ns for the PC_61_BM/MAPbI_3_ structure. Based on the TRPL results, the carrier lifetimes of the different structures demonstrate the benefits offered by the ZnO interface layer. They also show that the ZnO interface layer can prevent carrier recombination and shorten the carrier transmission time. Based on the results regarding the SEM images, PL spectra, and TRPL spectra, it was certified that the performance and structures of the PSCs could benefit from the use of an ZnO interface layer.

Figure 7a,b illustrate the current density–voltage and dark current density–voltage performance of the PSCs with PC_61_BM ETL formed using 50 mg/mL and ZnO interface layers of various thicknesses. The related performance values of the resulting PSCs are listed in Table 2. It was found that the performances of the PSCs were improved by inserting the ZnO interface layer. This was because the coverage of the ZnO interface layer could effectively passivate the fine cracks on the PC_61_BM surface and prevent carrier recombination. Moreover, the ZnO interface layer increased the degree to which the energy levels between the PC_61_BM ETL and the Ag cathode matched, thus improving the performance of the resulting PSCs. As shown in Figure 5, improved passivation features were exhibited when the 20 nm thick ZnO interface layer was used in comparison to the layer with a thickness of 10 nm. Consequently, the best performance was obtained when the 20 nm thick ZnO interface layer was used, yielding a PCE of 14.62%, a V_oc_ of 0.88 V, a J_sc_ of 22.57 mA/cm^2^, and an FF of 73.61%. However, as the thickness of the ZnO interface layer was increased to 30 nm, the PCE of the perovskite solar cells decreased to 14.42%. This was attributed to the fact that the series resistance (R_s_) of the PSCs with the 30 nm thick ZnO layer (as calculated from the J-V curve in Figure 7a) was increased from 6.02 Ω-cm^2^ to 6.30 Ω-cm^2^ in comparison to the PSCs with the 20 nm thick ZnO layer, which caused the deteriorating of carrier transmission. Figure 7c shows the EQE and integrated J_sc_ as a function of wavelength (300—800 nm) for the PSCs with the PC_61_BM ETL formed using 50 mg/mL and with ZnO interface layers of various thicknesses. The trend of EQE results also constituted a significant improvement from the above-mentioned J_sc_ measurement trend of the PSCs using the PC_61_BM ETL formed using 50 mg/mL and the ZnO interface layers of various thicknesses. This was attributed to the fact that the coverage of the ZnO interface layer and the series resistance affected the electron transport ability at the same time, which made the J_sc_ and EQE of the PSCs with the 20 nm thick ZnO interface layer larger than the one with the 10 nm thick and 30 nm thick ZnO interface layers. Moreover, the integrated J_sc_ values of the PSCs using the ZnO interface layer with thicknesses of 0, 10, 20, and 30 nm were 19.89, 20.33, 21.26, and 20.75 mA/cm^2^, respectively. The values and trends of the integrated J_sc_ of the PSCs using the ZnO interface layer with various thicknesses were all similar to those of the J_sc_ obtained from the J-V curve. The calculation results also proved that the optimal thickness of the ZnO interface layer was 20 nm. Thus, adding a ZnO interface layer with a suitable thickness can improve the performance of the resulting PSCs. The performance of the PSCs with a 20 nm thick ZnO interface layer was superior to that of the PSCs without a ZnO interface layer, and the associated PCE increased from 12.54% to 14.62%. The improved performance was attributed to the fact that the optimized ZnO interface layer could effectively retouch the PC_61_BM ETL surface to passivate the fine cracks and reduce the carrier recombination rate.

Finally, the reverse scan and forward scan of the J-V curve for the PSCs without and with a 20 nm thick ZnO interface layer were measured and shown in Figure 8. The device characteristics are listed in Table 3. According to the results, the V_oc_, J_sc_, and FF of the resulting PSCs were increased during the reverse scan. This was because the deficiencies between the layers would trap the ionic charges and enhance or deteriorate the built-in field (E_B_) during the reverse scan and forward scan, respectively, leading to the difference in the device performance [38]. Furthermore, based on the forward scan and reverse scan of the J-V curve for the solar cells, the hysteresis index (HI) of the device could be calculated, which represented the interface quality of the device. The hysteresis index was calculated using Formula (2), as follows:(2)$$Hysteresis\;index\left(HI\right)=\frac{{PCE}_{reverse}-{PCE}_{forward}}{{PCE}_{reverse}}$$
where PCE_reverse_ is the power conversion efficiency of the PSCs during a reverse scan (V ≥ V_oc_ to V ≤ 0), and PCE_forward_ is the PCE of the PSCs during a forward scan (V ≤ 0 to V ≥ V_oc_). In this work, along with the covering of the ZnO interface layer, the cracks and pinholes on the PC_61_BM ETL were retouched, and the interface quality was improved, thereby decreasing the hysteresis index. Compared with the PSCs without a ZnO interface layer, the hysteresis index of the PSCs using the 20 nm thick ZnO interface layer decreased from 7.52% to 4.88%. This result also verified the passivation function of the ZnO interface layer [39].

## 4. Conclusions

In summary, using various-concentration-formed PC_61_BM ETLs to cover the MAPbI_3_ active layers in the PSCs, it was found that the best performance could be obtained using the PC_61_BM ETL with a concentration of 50 mg/mL. According to the SEM images, the PC_61_BM ETL with the concentration of 50 mg/mL could uniformly and completely cover the MAPbI_3_ active layer and did not present cracks and pinholes on the surface. The PL and TRPL results also demonstrated that the PC_61_BM ETL with the concentration of 50 mg/mL had the best electron transportation ability and the lowest carrier recombination probability. However, the performance of the PSCs still suffered due to the appearance of some fine cracks on the PC_61_BM ETL surface. Therefore, to achieve further performance improvements of the PSCs by precluding the formation of fine cracks, the ZnO interface layer was deposited on the PC_61_BM ETL using a thermal evaporator. The additional ZnO interface layer not only effectively passivated the fine cracks on the surface but also improved the degree to which the energy levels between the PC_61_BM ETL and the Ag cathode matched, thus improving the performance of the PSCs. Consequently, the PSCs with the 20 nm thick ZnO interface layer presented the best performance because they had a superior passivation function compared with that of the PSCs with other ZnO interface layer thicknesses. The power conversion efficiency of the resulting PSCs increased from 12.54% to 14.62%. Moreover, the hysteresis index of the PSCs with the 20 nm thick ZnO interface layer decreased from 7.52% to 4.88% compared with that of the PSC without a ZnO interface layer. This finding proved that the covering of ZnO interface layers could indeed passivate the fine cracks on the PC_61_BM ETL and promote its carrier transportation ability. According to the experimental results, optimizing the PC_61_BM ETL and the ZnO interface layer can effectively improve the performance of PSCs.

## Figures and Tables

**Figure 1 materials-16-05061-f001:**
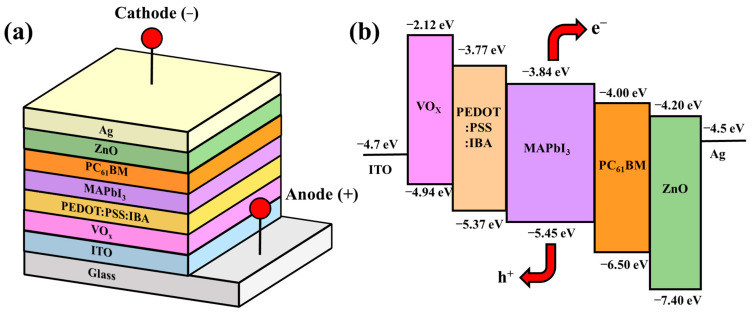
(**a**) Three-dimensional schematic configuration, (**b**) energy level diagram of perovskite solar cells with PC_61_BM electron transport layer and ZnO interface layer, and (**c**) SEM cross-section image of PSCs with 20 nm thick ZnO interface layer.

**Figure 2 materials-16-05061-f002:**
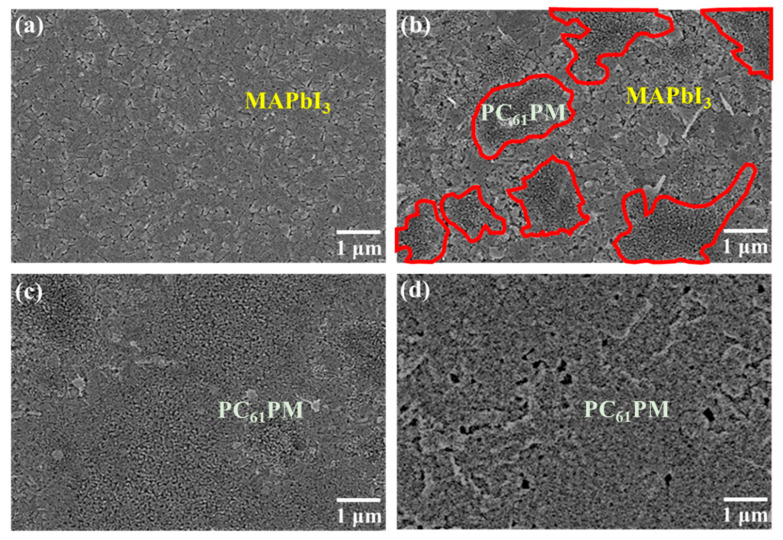
SEM images of (**a**) MAPbI_3_ films and PC_61_BM/MAPbI_3_ films with various PC_61_BM concentrations of (**b**) 30, (**c**) 50, and (**d**) 70 mg/mL.

**Figure 3 materials-16-05061-f003:**
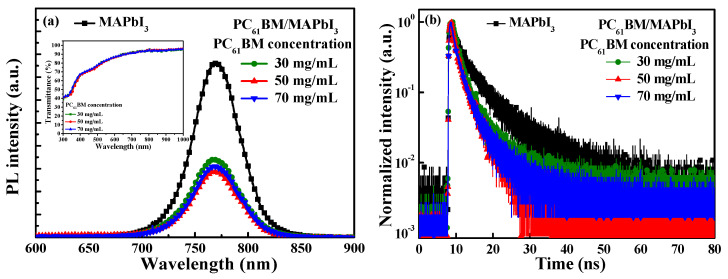
(**a**) PL spectra and (**b**) TRPL spectra of MAPbI_3_ active layer and various-concentration-formed PC_61_BM ETLs/MAPbI_3_ active layers. The inset figure in Figure 3a shows transmission spectra of various-concentration-formed PC_61_BM ETLs.

**Figure 4 materials-16-05061-f004:**
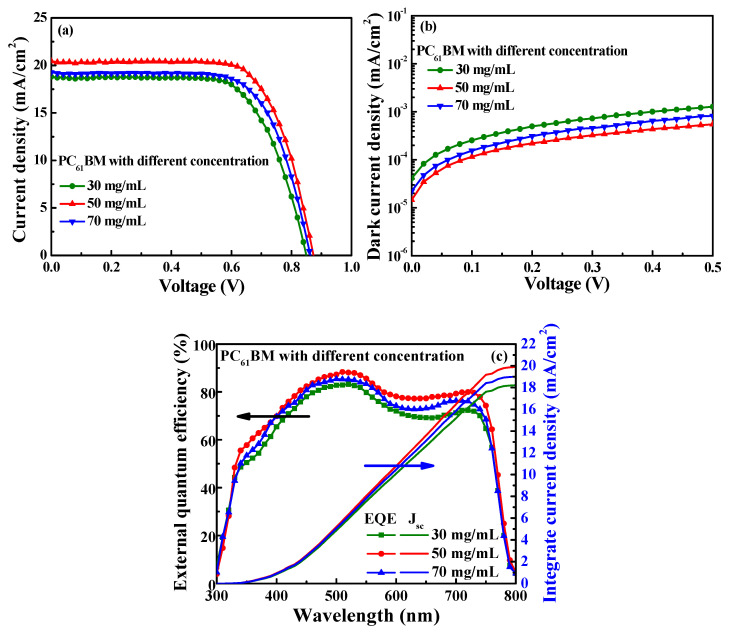
(**a**) Current density–voltage, (**b**) dark current density–voltage, and (**c**) external quantum efficiency spectra and integrated J_sc_ characteristics of PSCs with PC_61_BM ETL formed using various PC_61_BM concentrations.

**Figure 5 materials-16-05061-f005:**
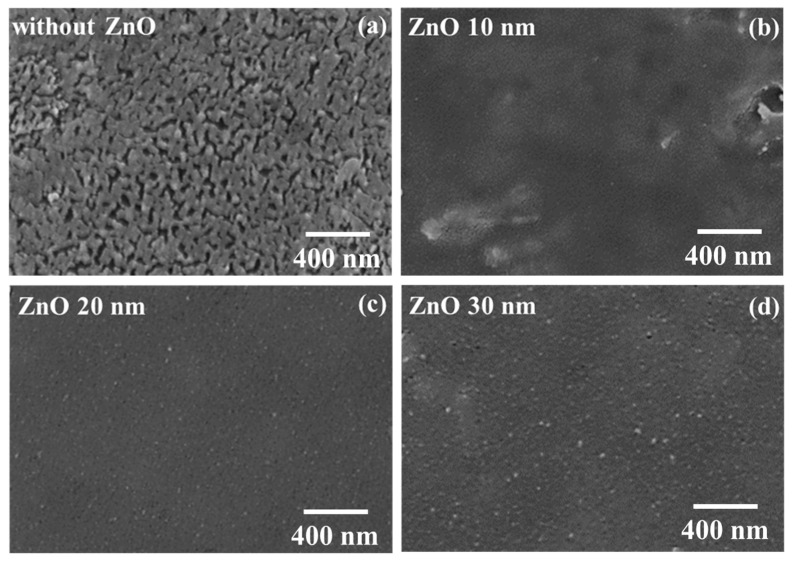
SEM images of (**a**) PC_61_BM/MAPbI_3_ and ZnO/PC_61_BM/MAPbI_3_ with various ZnO thicknesses of (**b**) 10, (**c**) 20, and (**d**) 30 nm.

**Figure 6 materials-16-05061-f006:**
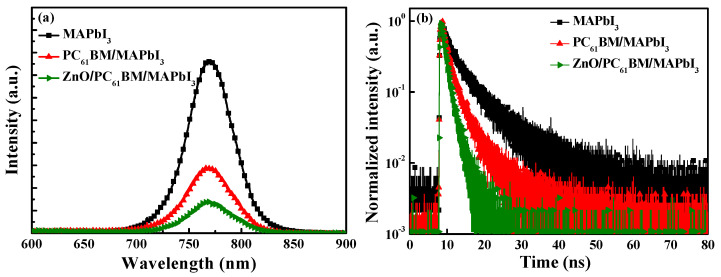
(**a**) PL spectra and (**b**) TRPL spectra of MAPbI_3_, PC_61_BM/MAPbI_3_, and 20 nm thick ZnO/PC_61_BM/MAPbI_3_ structures.

**Figure 7 materials-16-05061-f007:**
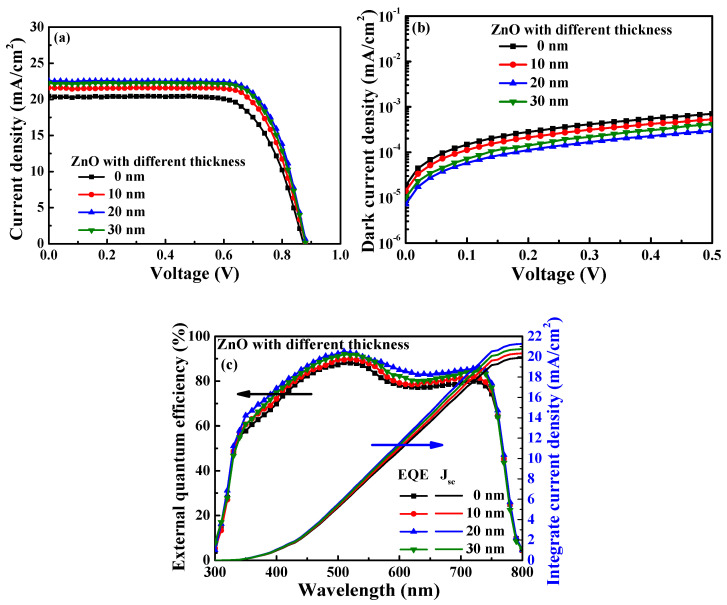
(**a**) Current density–voltage, (**b**) dark current density–voltage, and (**c**) external quantum efficiency and integrated J_sc_ characteristics of PSCs with various ZnO interface layer thicknesses.

**Figure 8 materials-16-05061-f008:**
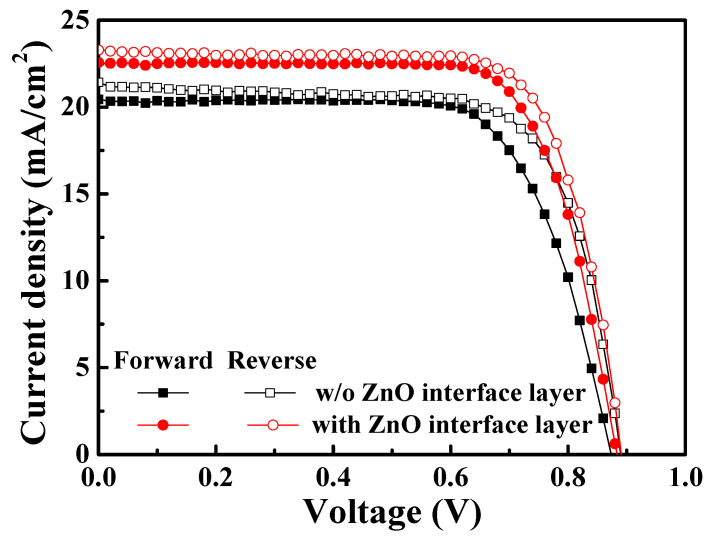
J-V curve during reverse scan and forward scan of PSCs without and with 20 nm thick ZnO interface layer.

**Table 1 materials-16-05061-t001:** Characteristics of PSCs with PC_61_BM ETL formed using various PC_61_BM concentrations.

PC_61_BM Concentration (mg/mL)	V_oc_ (V)	J_sc_ (mA/cm^2^)	FF (%)	PCE (%)	Integrated J_sc_ (mA/cm^2^)
30	0.85	18.79	67.81	10.83	18.22
50	0.87	20.44	70.52	12.54	19.89
70	0.86	19.28	69.48	11.52	19.00

**Table 2 materials-16-05061-t002:** Characteristics of PSCs with various ZnO interface layer thicknesses.

ZnO Thickness (nm)	V_oc_ (V)	J_sc_ (mA/cm^2^)	FF (%)	PCE (%)	Integrated J_sc_ (mA/cm^2^)
0	0.87	20.44	70.52	12.54	19.89
10	0.88	21.62	72.74	13.84	20.33
20	0.88	22.57	73.61	14.62	21.26
30	0.88	22.29	73.51	14.42	20.75

**Table 3 materials-16-05061-t003:** Characteristics and hysteresis index of PSCs without and with 20 nm thick ZnO interface layer during reverse scan and forward scan.

Structure (Scan Direction)	V_oc_ (V)	J_sc_ (mA/cm^2^)	FF (%)	PCE (%)	Hysteresis Index (%)
Without ZnO (Forward)	0.87	20.44	70.52	12.54	7.52
Without ZnO (Reverse)	0.89	21.43	71.10	13.56	
With 20 nm ZnO (Forward)	0.88	22.57	73.61	14.62	4.88
With 20 nm ZnO (Reverse)	0.89	23.29	74.14	15.37	

## Data Availability

The data presented in this study are available on request from the corresponding author.

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
