# Peer review of "Investigation of the Performance of Perovskite Solar Cells with ZnO-Covered PC61BM Electron Transport Layer"

_materials, 2023, doi:10.3390/ma16145061_

Round 1
Reviewer 1 Report
This manuscript presents a study of PSCs with original architecture, including PC61BM as an ETL and ZnO as an interface layer. The study was carried out qualitatively and the results obtained are presented clearly. Therefore, this work can be recommended for publication in a journal. However, the following questions should be resolved to better understanding results:
1. Has the interaction between PC61BM and ZnO been studied in any way other than assessing the morphology of the layers. This is where the XPS study can be helpful.
2. In Figure 4с and 7c, the photocurrent density values from IPCE spectra were calculated. Please compare them with the Jsc values from J-V curves. They are important information that guarantees the validity of the photovoltaic measurements.
3. In Table 1 and 2, how many devices the authors fabricated for obtaining the average cell performances? Please provide the statistical analysis.
4. If possible, I would like that the authors to add forward and reverse scan data and stability data for the best device.
Reviewer 2 Report
Dear Authors,
I would like to inform you that I have read your manuscript and I have found it interesting and well prepared. All results are presented clearly; we can find both the good quality of figures and their detail descriptions and the suitable conclusions and references.
However, I have only these remarks:
- - How were obtained thicknesses of individual layers of solar cell ?
- - Transmission spectra should be presented in the spectral range from 300 nm, to show the absorption of a He-Cd laser source with a wavelength of 325 nm (the excitation spectra should be presented, not only PL emission).
Taking into consideration all above mentioned comments I accept your manuscript for publications in MATERIALS, after minor revision.
Reviewer 4 Report
In this manuscript, the authors studied the ZnO2 covered PC61BM electron transport layer to enhance the performance of the perovskite solar cells. Using the 30-nm-thick PC61BM ETL with a concentration of 50 mg/mL, the performances of the PSCs were the open-circuit voltage (Voc) of 0.87 V, short-circuit current density (Jsc) of 20.44 mA/cm2, fill factor (FF) of 70.52%, and power conversion efficiency (PCE) of 12.54%. The manuscript should be accepted after the following modifications;
1) XRD is the basic tool to study the materials. The authors should add XRD data of the samples and add more information’s such as lattice parameters etc.
1) In Experimental section; the authors mentioned the thickness of each layer; the author should provide cross-sectional image the device to confirm the thickness
2) What is the value of contact angle; did the authors check it?
3) The interface among the each layer is should be clear to obtain a high performance device; what’s about the interface quality.
4) Did the authors check the hysteresis index (HI) of the device; the authors should measure the device in forward and reverse scan and measure the HI
5) The ideality factor n can be a key parameter to estimate the suppression of the recombination rate in perovskite device. The authors should measure the dark current and n
6) On page 11, the authors mentioned that the carrier recombination probability was reduced and the lowest dark current density was resulted…………..” the reduction of charge recombination of the sample device should be analysed further on the basis of the open-circuit voltage Voc because all of the photo-excited charge carriers will recombine within the device at the end at the open-circuit condition.
7) The surface of the thin films play a vital role in the absorbance and transmittance. The authors measured the roughness of each films and compared the results using AFM
8) There are some recent and relevant papers that should be cited; doi.org/10.1038/s41598-020-80640-3; doi.org/10.1016/j.jallcom.2022.166007
9) Many spelling and formatting typos in this paper and the authors should check and revise them thoroughly.
Many spelling and formatting typos in this paper and the authors should check and revise them thoroughly.
Round 2
Reviewer 4 Report
Accepted in the present form.
the authors have improved the language
